# Heterogeneous dissociation process of truncated RNAs by oligomerized Vasa helicase

Yoshimi Kinoshita[1], Ryo Murakami[1], Nao Muto[1], Shintaroh Kubo[2], Ryo Iizuka [1] & Sotaro Uemura [1✉]

RNA helicases are enzymes that generally unwind double-stranded RNA using ATP hydrolysis energy, mainly involved in RNA metabolism, transcription, translation, and mRNA splicing. While the helicase core is crucial for RNA unwinding activity, N- and C-terminal extensions of specific helicases may contain an intrinsically disordered region for electrostatic interaction, resulting in the formation of droplets in the cytoplasm. However, how the disordered region of the RNA helicase contributes to RNA unwinding and dissociation remains unclear. Here, we focused on *Bombyx mori* Vasa, which unwinds truncated target transposon RNAs from the piRNA-induced silencing complex piRISC. In this study, we used single-molecule techniques to visualise how Vasa dynamically interacts with piRISC and investigate how Vasa oligomerization is involved in the process of piRNA amplification, named the ping-pong pathway. We found that Vasa's oligomerization is required during these processes in vitro and in vivo, and that Vasa triggers the dissociation of truncated RNA in heterogeneous pathways. Our single-molecule results suggest that oligomerized Vasa guides the timing of the process regulating overall dissociation efficiency.

[1] Department of Biological Sciences, Graduate School of Science, The University of Tokyo, Tokyo, Japan. [2] Department of Biophysics, Graduate School of Science, Kyoto University, Kyoto, Japan. ✉email: uemura@bs.s.u-tokyo.ac.jp

D EAD-box RNA helicases are the largest helicase family involved in all aspects of RNA metabolism, including transcription, translation, and mRNA splicing[1]. These RNA helicases are enzymes that typically unwind double-stranded RNA using the hydrolysis energy of ATP in highly conserved residues, including the D-E-A-D motif[1–3]. RNA helicases disrupt the conformations of elongated RNAs with specific secondary structures and reconstruct the RNA-protein complex by the ATP-driven facilitation of the helicase core to the closed conformation, consisting of two RecA-like domains with flanking linkers[4,5]. While the helicase core is crucial for RNA unwinding, the N- and C-terminal extensions of specific helicases have a unique function. For example, the N-terminus of the human homolog of DEAD-box helicase 4 (DDX4) may contain an intrinsically disordered region (IDR) for electrostatic interaction with other DDX4 molecules, resulting in the formation of droplets in the cytoplasm[6]. These results suggest that the N-terminus region of DDX4 plays an important role for aggregation in cells. Thus, the amino-acid sequence of Vasa, a homolog of human DDX4, suggests that its N-terminal IDR undergoes oligomerization. However, the mechanism by which Vasa oligomerization links to the unwinding function of RNA is yet unknown.

Therefore, in this study, we focused on the function of the N-terminal region of *Bombyx mori* Vasa, an ATP-dependent RNA helicase that unwinds truncated target transposon RNAs from the piRNA-induced silencing complex (piRISC) during piRNA metabolic processes[7,8]. This cleavage reaction is induced by the formation of piRISC by piRNA (PIWI-interacting RNA), a noncoding RNA generated specifically for germ cells, and Siwi (Silkworm Piwi), one of the PIWI (P-element induced wimpy testis) proteins of the Argonaute subfamily (AGO and PIWI proteins). Furthermore, germ cells have an ingenious piRNA amplification mechanism for efficient transposon RNA cleavage, known as the ping-pong pathway[9]. In this pathway, Siwi, Vasa, Ago3, and DDX43 are orchestrate a droplet called Nuage near the nuclear membrane in silkworm germ cells BmN4[8,10].

Although dynamic changes in truncated RNA dissociation triggered by Vasa in piRISCs are intimately tied to the ping-pong pathway, the mechanism by which Vasa interacts with piRISC and triggers truncated RNA dissociation remains unknown. In this study, we used single-molecule techniques to visualise how Vasa dynamically interacts with piRISC and evaluate how Vasa's oligomerization is involved in the ping-pong pathway. As the result, we found that Vasa triggers the dissociation of truncated RNA in heterogeneous pathways, and that Vasa oligomerization is required during these processes both in vitro and in vivo.

## Results

**N-terminal IDR contribution for Vasa's dimerization**. To determine whether Vasa undergoes oligomerization at the N-terminal domain, both wild-type ("WT-Vasa") and a mutant with N-IDR deletion ("dN-Vasa") were prepared using anti-FLAG gel column by culturing germ cell line BmN4 derived from *Bombyx mori* (silkworm) ovary (Fig. 1a). Both samples were fluorescently labelled through a Halo-tag and stably interacted with TMR-labelled Halo ligands. First, a distribution of the fluorescence intensity of the fluorescently labelled Vasa on the glass surface was obtained. Although WT-Vasa mostly formed dimers, over 90% of dN-Vasa formed monomers based on the distribution of the fluorescence intensity (black and grey bars in Figs. 1b, S1a–b), indicating that Vasa dimerizes through intermolecular interactions through N-IDR.

Next, we visualised the oligomerized Vasa in vivo. BmN4 cell imaging showed that the transfected WT-Vasa formed Nuage, whereas dN-Vasa did not form Nuage with dispersion in the cytoplasm (Fig. 1c), suggesting Vasa's dimerization at N-IDR triggers Nuage formation.

**Aggregation dynamics of association and dissociation**. Since Vasa has RNA-binding recognition sites other than IDRs[4,7], Vasa has been found to aggregate more in the presence of exogenous RNA using TMR-labelled Vasa. To observe the effect of exogeneous RNA on Vasa's oligomerization, biotinylated TMR-labelled Vasa was immobilised on a glass substrate at a single-molecule density in the presence of yeast RNA (stripe bars in Fig. 1b). This result indicates that exogenous RNA contributes to the further oligomerization of Vasa dimerized through IDR.

Next, we observed the real-time dissociation and association process in the presence of 50 nM TMR-Vasa (Fig. 1d). The number of Vasa molecules in association and dissociation was calculated to be $2.13 \pm 0.17$ and $2.48 \pm 0.25$, respectively (Fig. 1e), indicating that the dynamics of aggregation occurred in the dimer unit. The dwell time for Vasa to associate or dissociate was constant, $465.3 \pm 34.4$ s and $385.1 \pm 15.3$ s, respectively (Fig. 1f), independent of the number of Vasa molecules (Fig. S1c). This indicates that the allosteric behaviour, in which the dynamics of aggregation are modulated by the number of Vasa molecules, was not observed. These results indicate that deletion of the IDR abrogates dimerization as well as Nuage formation and RNA unwinding activity. The simplest interpretation is indeed that dimerization is required for these activities. However, it cannot be ruled out that the IDR has functions other than promoting dimerization that are essential for these activities.

**Both ATP and oligomerization dependent unwinding activity**. Next, to reveal how the oligomerization of Vasa directly influences the unwinding activity of target cleaved RNA, we established a reconstitution system using Siwi-piRISC, Vasa, and target RNA fluorescently labelled with ATTO647N at the 5′ end, to visualise the unwinding process using total internal reflection fluorescence (TIRF) microscopy at the single-molecule level. We first confirmed the cleavage and unwinding activities of the reconstructed system in the bulk. The results showed that in the presence of WT-Vasa, target RNA complementary to endogenous piRNA was dissociated from the beads bound to Siwi-piRISC (Fig. 2a). On the other hand, in the case of dN-Vasa, the amount of cleaved and dissociated RNA fragments was lower than in the case of WT-Vasa, demonstrating that only dimer units, not monomers, have unwinding activity.

The 3′ end of target RNA in Siwi-piRISC was immobilised on the glass surface via a biotin-avidin bond at higher densities. The average fluorescence intensity of target RNA in the field of view was monitored in real time immediately after the introduction of non-fluorescent WT-Vasa and dN-Vasa (Fig. 2b). In the absence of ATP, the fluorescence signal remained almost constant over time, while in the presence of 1 mM ATP, it decayed dramatically within a few minutes (Figs. 2c and S2). This suggests that ATP hydrolysis by Vasa induces the dissociation of target RNAs after cleavage by piRISC. On the other hand, in the case of dN-Vasa, almost no signal decay was observed even in the presence of ATP, demonstrating that the target RNA was only negligibly dissociated (Fig. 2c). These results were consistent with the results of the cleavage assay in bulk, further confirming the effect of ATP hydrolysis.

We subsequently monitored the individual dissociation processes that were not obtained from the averaged signals. By reducing the concentrations of immobilised target RNA to 10–100 pM, fluorescent signals could be successfully identified at single-molecule resolution (Fig. 2d). Since the unwinding event was expected to be dependent on the ATP hydrolysis process, the

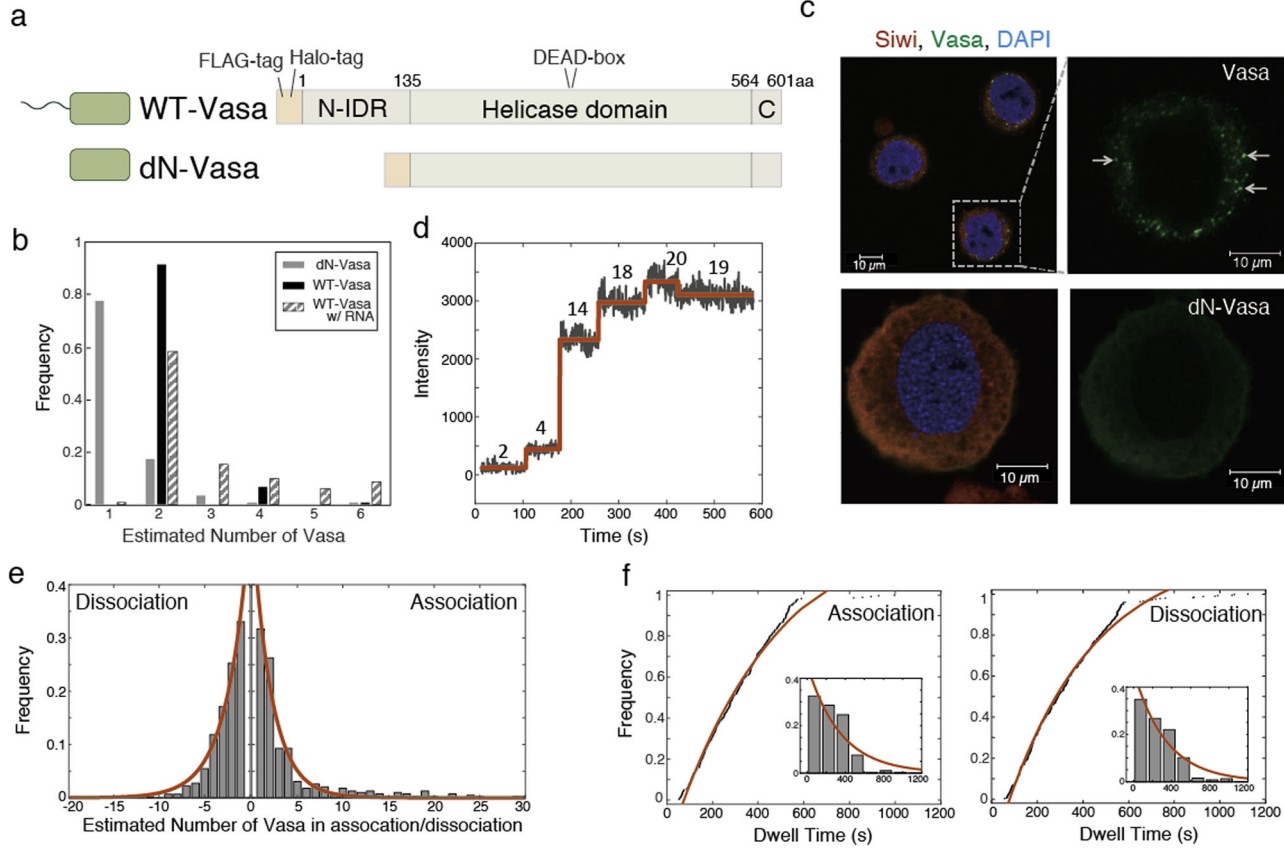

**Fig. 1 The role of N-terminus in Vasa for Vasa-to-Vasa oligomerization process. a** Schematic sequence of recombinant Vasa and dN-Vasa. N-IDR and C-terminal regions are disordered. DEAD-box is in the helicase domain. **b** The distributions of the estimated number of Vasa molecules at a single spot of WT-Vasa (black, total 198 spots) and dN-Vasa (grey, total 113 spots). Stripe bars indicate the distribution of WT-Vasa in the presence of RNA (total 140 spots, see "Methods"). The molecular numbers of Vasa were calculated from the averaged intensity of a single TMR molecule (**b**, **d**, **e**). **c** Immunostaining of BmN4 cells. Arrows show the nuage where Vasa aggregate. Scale bars represent 10 μm. **d** An example trajectory of fluorescence intensity of Vasa. Numbers indicate the molecular number of Vasa calculated from the fluorescence intensity. **e** The distribution of the estimated number of Vasa molecules in association and dissociation. Association and dissociation molecules were calculated to be 2.13 ± 0.17 ($R^2$ = 0.964, 388 events of 173 spots) and 2.48 ± 0.25 ($R^2$ = 0.987, 439 events), respectively. **f** The cumulative distribution of the dwell time before association and dissociation. Time constants in association and dissociation were calculated to be 465.3 ± 34.4 s ($R^2$ = 0.986, 282 events) and 385.1 ± 15.3 s ($R^2$ = 0.991, 282 events), respectively. Insets show the corresponding histograms fitted with a single exponential curve.

dwell time to the disappearance of the fluorescence signal in the presence of low and high concentrations of ATP was compared. As a result, the dwell time until the dissociation of the cleaved target RNA was found to be five times longer in the presence of 10 μM ATP than in the presence of 1 mM ATP (Fig. 2e). These results indicate that Vasa as an ATP-driven RNA helicase is directly responsible for the dissociation of target RNA post-cleaved by Siwi-piRISC even at the single-molecule level.

**Colocalization of oligomerized Vasa and RNA.** Furthermore, to visualise how Vasa directly interacts with and dissociates from Siwi-piRISC in the presence of exogenous piRNA-4 (see "Methods"), we simultaneously measured the behaviour of Vasa and target RNA fluorescently labelled with TMR and ATTO647N, respectively. The results showed that Vasa colocalized well in the presence of Siwi-piRISC, but not in its absence in TIRF experiments (Figs. 3a, S3c). This indicates that the target RNA and Vasa form a complex via Siwi-piRISC. Vasa colocalized with Siwi-piRISC was found to be oligomerized over a wide range from 2 to 10 molecules (Fig. 3b), suggesting that Vasa's oligomerization is enhanced by the formation of a complex with Siwi-piRISC. In contrast, dN-Vasa hardly colocalized with the target RNA,

indicating that the IDR-mediated dimerization of Vasa contributes to the oligomerization induced by Siwi-piRISC formation.

In addition, we also visualised Siwi via Siwi antibody fluorescently labelled with Alexa 488 and confirmed the simultaneous localisation of the three factors including Vasa and target RNA (Fig. S4). This indicates that the target RNA, Vasa, and Siwi form a stable complex at the single-molecule level in the absence of ATP.

**Heterogeneous pathways in the dissociation of RNA and Vasa.** Next, we observed the dissociation process of Vasa and target RNA from the Siwi-piRISC complex in the presence of ATP. Both dissociated simultaneously in 19% of traces (N = 14 out of 72 traces in total) (Figs. 3c and S5a), while Vasa and target RNA dissociated at different times in 44% of traces (N = 31 out of 72 traces) (Figs. 3d and S5b).

In the case of dissociation at different times, Vasa progressively dissociated from the oligomerized Vasa. Furthermore, even after the RNA dissociated, a part of Vasa remained bound to the truncated RNA immobilised at the 3′ end on the surface. This suggests that a part of the Vasa remains bound to the truncated

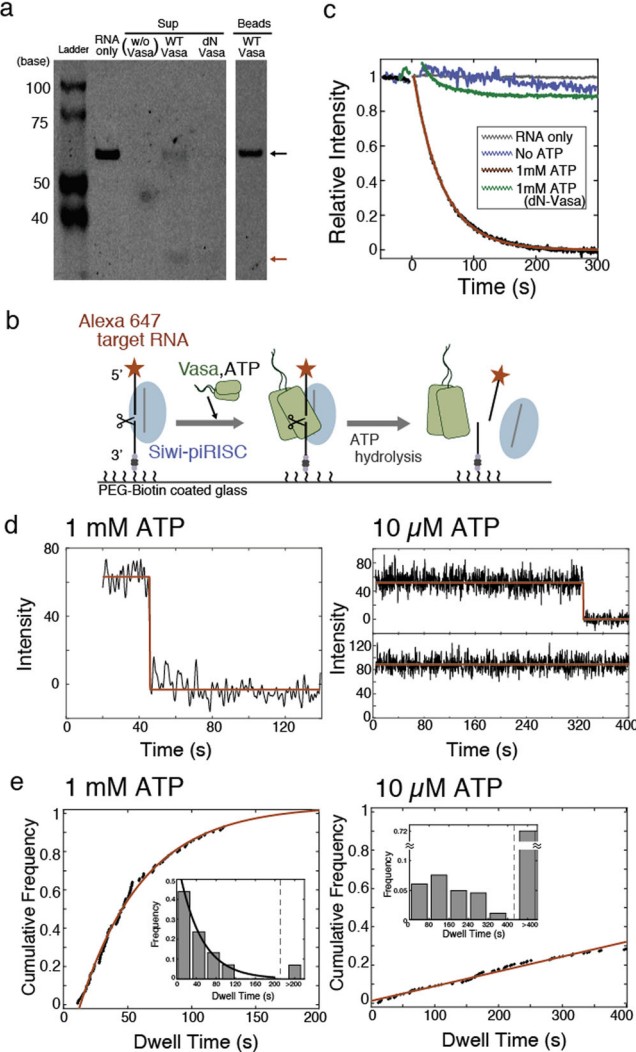

**Fig. 2 RNA cleavage assay by Siwi-piRISC immobilized on the surface.**
**a** RNA cleavage assay of target RNA. In the presence of Siwi, Vasa and ATP, target RNA was cleaved and dissociated from Siwi-piRISC in bulk. Black and red arrows indicate the target RNA (64 nt) and cleaved RNA (31 nt), respectively. Full gel image is shown in Fig. S6a. **b** Schematic representation of dissociation of cleaved target RNA loaded on Siwi-piRISC and Vasa. **c** The relative intensity decay of target RNA after adding 0 (blue) or 1 mM (black) ATP with Vasa in a certain (27.6 × 27.6 μm²) region. The red curve shows the exponential decay with the time constant of 46.5 ± 0.2 s ($R^2 = 0.998$) in addition of 1 mM ATP. With dN-Vasa, the intensity was shown in green. The intensity of fluorescent-labelled RNA was constant coloured in grey. **d** The example time-trace intensity of single target RNA after adding 1 mM or 10 μM ATP. Red lines show the analysed stepwise intensity. **e** The distribution of dwell time before cleaved RNA detached in 1 mM ATP with the time constant of 49.8 ± 2.9 s ($R^2 = 0.994$) or in 10 μM ATP. 71.6% of RNA remained over 400 s in 10 μM ATP. Insets are the corresponding histograms fitted with a single exponential curve.

RNA even after the dissociation of the 5′-terminal truncated RNA attached to ATTO647N.

In the remaining 37% of traces, the target RNA did not dissociate within 400 s ($N = 27$) (Figs. 3e and S5c). Although it is not possible to determine how often these cases actually occurred, there are three possibilities in this situation. The first is that dissociation occurred after 400 s. The second is that Siwi cleavage does not occur because piRNA and target RNA are not complementary, and Vasa unwinding is not observed. Lastly,

the third possibility is that Siwi cleavage occurs, but unwinding is not observed.

In all cases, oligomerized Vasa progressively dissociated independently of the dissociation event of the target RNA (Figs. 3d–f, S5a, b). In addition, the partial dissociation of Vasa was observed almost at the single-molecule level (Fig. 3g).

In summary, the obtained results suggest that these heterogeneous dissociations may contribute to the overall RNA silencing activity of the ping-pong cycle.

## Discussion

Our single-molecule data demonstrated the significance of oligomerization for DEAD-box RNA helicase, Vasa, to trigger the dissociation of truncated RNA using ATP hydrolysis energy. Recently, Andrea et al. reported that one of the DEAD-box helicases, yeast Ded1p or human ortholog DDX3, forms oligomers and modulates unwinding activity in cells and in vitro[11]. The RNA-binding and unwinding activity and IDRs of the *Drosophila* ortholog Belle are also required for viability and fertility[12,13]. Furthermore, in addition to DEAD-box RNA helicase, one of the prokaryotic DNA helicases, *Escherichia coli* UvrD forms oligomers to unwind duplex DNA, as indicated by not only single-molecule imaging approaches[14] but also biochemical approaches[15]. Therefore, the Vasa oligomerization observed in this study is consistent with previously reported experimental results. Our further coarse-grained MD simulation supported the importance of IDRs for Vasa oligomerization. (Fig. S1d–f; see Supporting Information). We have not performed experiments using the mutant with different truncated regions. By focusing on the repetitive regions in the IDRs, experiments using various truncated regions will reveal more details about the required regions for the dimerization.

The characteristic interactions of oligomerized Vasa with a single Siwi-piRISC were verified. We observed multiple pathways for the dissociation of a single truncated RNA induced by ATP-dependent Vasa for the first time (Figs. 3c–f, 4). Our results suggest that oligomeric Vasa with a minimal unit of dimer associates with Siwi-piRISC in oligomeric form through IDR interaction, and recognises the cleaved RNA, leading to the dissociation process. In the ping-pong model, Ago3 is required to incorporate the RNA fragment truncated and dissociated by Vasa as a new piRNA. Therefore, the formation of oligomers of Vasa followed by the formation of Nuage in the cell may play an important role in the delivery of truncated RNA to Ago3. RNA cleavage, dissociation and incorporation are efficiently carried out at a physiological concentration of both RNA and helicase in the nuage, forming phase-separated granules in cells. To understand the functions of Vasa beyond RNA helicase activity, further single-molecule analysis should be performed for the simultaneous visualisation of cleaved RNA, Ago3, and Vasa.

In addition, our single-molecule trucking revealed heterogeneous pathways on dissociation processes between truncated RNA and Vasa, demonstrating that Vasa guides the timing of the process to regulate the overall dissociation efficiency, albeit via unclear mechanisms.

Single-molecule fluorescence methods have recently probed the dynamics during the RNA helicase unwinding process. For example, the siRNA functions of RNA helicase-independent human or Drosophila Ago2 have been visualised using single-molecule measurements[16–18]. Consistent with our results, heterogeneous pathways have been proposed and are significant in the RNA unwinding process. Our results revealed that oligomeric helicase is required for piRNA function and is expected to help make further advances in the elucidation of complicated piRNA regulation systems.

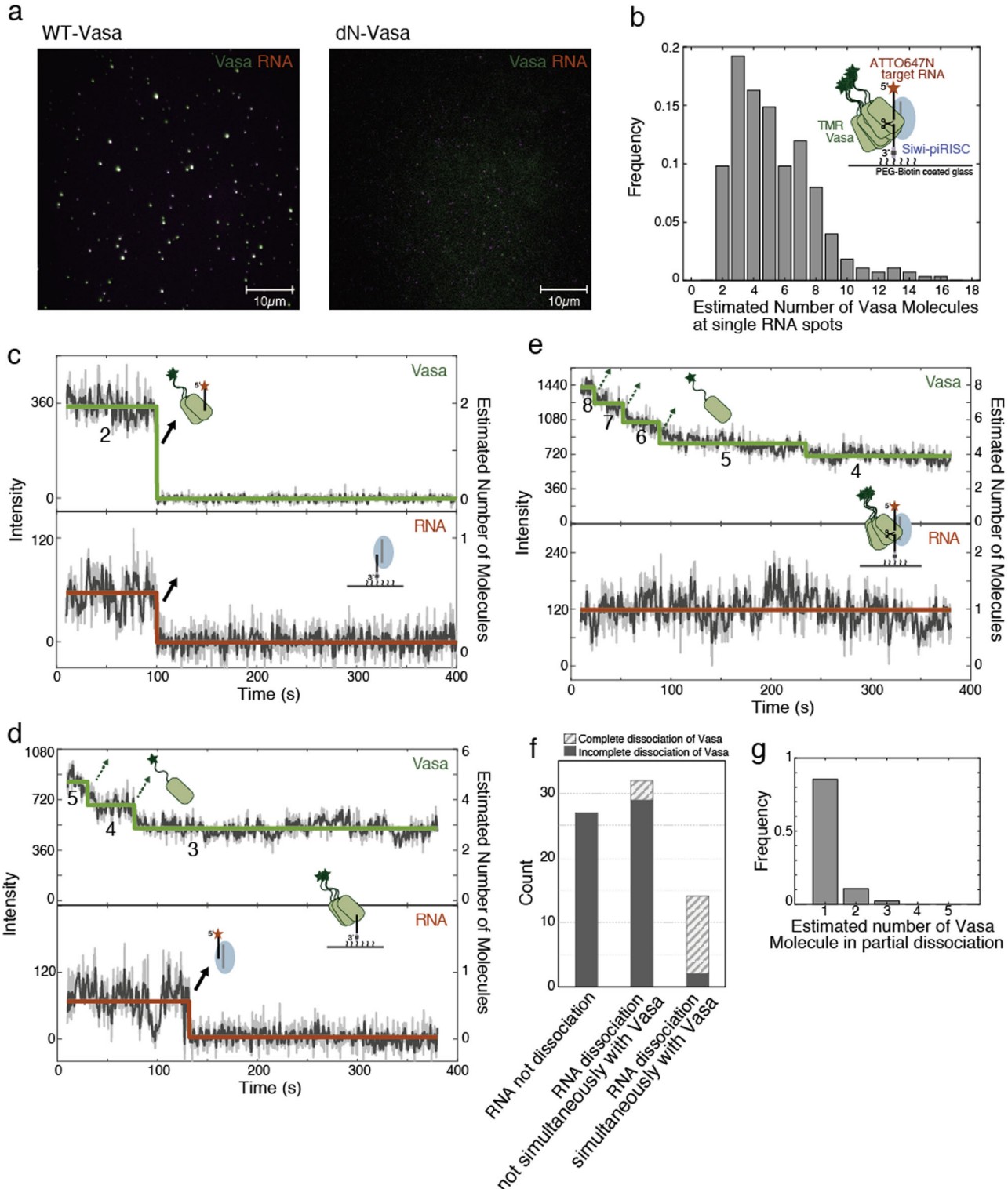

**Fig. 3 Single-molecule simultaneous tracking of target RNA and Vasa dissociation. a** The fluorescent images of target RNA (magenta) colocalized with the oligomerized WT-Vasa or dN-Vasa (green) in the presence of Siwi. Scale bars represent 10 μm. **b** The distribution of the number of TMR-Vasa molecules associated with a single molecule of target RNA, calculated from the fluorescence intensity ($N = 276$). The molecular numbers of Vasa and RNA were calculated from the averaged intensity of the respective single fluorescent dye (**b–e**, **g**). Inset shows the schematic representation of the complex with target RNA and Vasa loaded on Siwi-piRISC. **c–e** Time-trace intensities of the colocalized Vasa (upper, green) and cleaved RNA (lower, red) dissociating after adding 1 mM ATP. The intensity step was analysed, and the change in the molecular number of each factor converted from the fluorescence intensity was illustrated. **c** Vasa and RNA dissociated simultaneously. **d** Vasa partially dissociated independently of RNA dissociation. **e** Vasa partially dissociated but RNA kept binding. **f** Classify the dissociation of Vasa and RNA according to (**c–e**). The counts of Vasa dissociation completely are given in brackets. **g** Vasa dissociated partially as a single molecule (85%) in the colocalized spots (96 data in 52 spots).

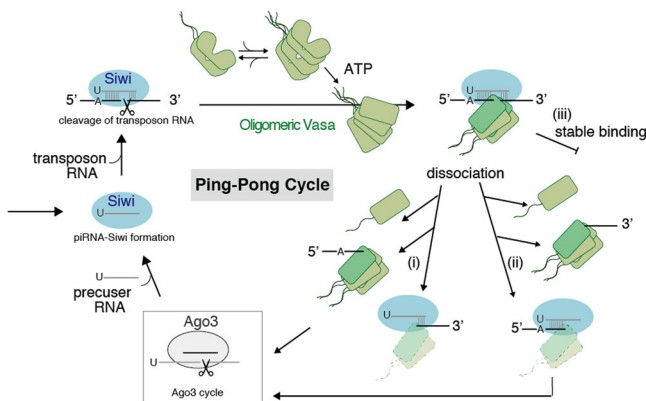

**Fig. 4 A multi-branched ping-pong cycle.** Oligomeric Vasa shown in green binds to Siwi-piRISC shown in blue in the upper left. Next, it undergoes the following three dissociation processes. (i) Oligomeric Vasa dissociates with truncated RNA at the 5′ end (ii) oligomeric Vasa dissociates with truncated RNA at the 3′ end, and (iii) oligomeric Vasa remains bound in a stable state. Thereafter, truncated RNAs would be transferred to Ago3, resulting in the formation of Siwi-piRISC again.

Finally, in the ping-pong model, DDX43, one of DEAD-box RNA helicase recently identified, was reported to dissociate truncated RNA from Ago3-piRISC[10]. In contrast, Vasa is responsible for dissociating RNA from Siwi-piRISC. Interestingly, DDX43 and Vasa share distinct roles. A more detailed analysis of these roles will provide clues as to how the ping-pong cycle acquired an efficient mechanism for transposon cleavage.

## Methods

**Design of artificial target RNA and piRNA-4.** The endogenous target RNA, 5′-CGA CCU CGA GGG GAG UCC AGA UUU GAA UCC GUU AGA UUA CAA UCA AGC UUA UCG AUA CCG UCG A-3′ (70 nucleotides, underline; complementary of endogenous piRNA, GeneDesign Inc.) was designed. The 5′ end of the target RNA was fluorescently labelled with Alexa647 via the SSH amino linker, and the 3′ end was labelled with biotin.

We also designed an exogenous target RNA ("piRNA-4 target") complementary to piRNA-4.

The sequence of piRNA-4 (28 nt) was 5′-UAA AUU CCC AGA AAC CGA AGU UAC UCC C-3′, where the 5′ end was phosphorylated and the 3′ end base (C) was methylated[8]. The designed sequence of target RNA (called "piRNA-4 target RNA") was 5′-CCC CCG GGA GUA ACU UCG GUU UCU GGG AAU UUA AAG CUU UAC AAU AAU ACC GUC GAC CUC GUG UAC UGG C-3′ (70 nucleotides, underline: complementary of piRNA4, GeneDesign Inc.). The 5′ end was fluorescently labelled with ATTO647N via an SSH amino linker, and the 3′ end was labelled with biotin.

**Preparation of protein constructs.** FLAG-Siwi and FLAG-Halo-Vasa were expressed in BmN4 cells, a germ cell line derived from *Bombyx mori* (silkworm) ovary, and purified using an anti-FLAG gel column (Sigma-Aldrich, #A2220). In the case of Vasa, we also prepared deleted constructs of the N-terminal disordered region (K133-W601 referred to as dN-Vasa). FLAG-tag and HaloTag (Promega) were fused at the N-terminus of Siwi and Vasa (Fig. 1a). HaloTag was fluorescently labelled with TMR-labelled HaloTag® Ligands (Promega) by incubating the optimal ligand to BmVasa 35 times for 2 h at room temperature, and then excluding free ligand by centrifugation (Micro Bio-Spin™ 6 column; Bio-Rad). The labelling rate was almost 100 %, as determined by spectrometry.

**Immunostaining.** Immunostaining experiments were performed after the transfection of Vasa or dN-Vasa into BmN4 cells, followed by incubation at 26 °C for 3 days. BmN4 cells were fixed in 4% formaldehyde for 15 min at room temperature. After permeabilization with 0.1% Triton X-100, the cells were washed with 3% w/v BSA in PBS. Vasa was treated with anti-FLAG M2 antibody (IgG1; Sigma-Aldrich, 1000-fold dilution) as the primary antibody while Siwi was treated with anti-Siwi antibodies (IgG2a, 1000-fold dilution) kindly gifted from Siomi lab at the University of Tokyo, followed by treatment with a secondary antibody, IgG1/IgG2a labelled with Alexa Fluor 488/555 (Invitrogen, 1000-fold dilution), for fluorescent detection. The nuclei were stained with DAPI. Fluorescent images were obtained using an LSM800 with AiryScan (Zeiss).

**Cleavage assay.** BmN4 lysates were prepared and incubated with Dynabeads Protein G (Invitrogen) and anti-Siwi antibody (2 µg) for 1 h at 4 °C. Guide RNA (500 nM, piRNA-4) was first added to the beads and incubated for 30 min at 26 °C, as necessary. Fluorescently labelled target RNA (50 nM) was then added to the beads and incubated for 3 h at 26 °C in cleavage buffer (25 mM HEPES [pH 7.3], 100 mM potassium acetate, 2 mM magnesium acetate, 5 mM DTT, 4 µg/ml yeast RNA, and 40 U/ml RNase inhibitor). Then, 500 nM Vasa and 1 mM ATP were added to the beads and incubated in cleavage buffer for 2 h at 26 °C. After the supernatant and beads were collected separately, RNAs in these fractions were extracted by phenol-chloroform treatment and precipitated with ethanol. Electrophoresis was performed using denaturing gel in 15% urea, and the fluorescence of RNA in the gel was scanned with Typhoon FLA 9500 (GE Healthcare).

**Fluorescent measurement of oligomerized Vasa using TIRF microscope.** Cover glasses were cleaned using 1 N KOH sonication and a plasma cleaner, and then the micro-chamber was built using adhesive tape[19]. The observed glass surface of the microchamber was coated with 0.01% poly-L-lysine for 5 min. The carboxylic acid group in Vasa was strongly adsorbed onto the glass surface in the assay buffer containing the oxygen scavengers: 30 mM HEPES, pH 7.3, 150 mM potassium acetate, 5 mM magnesium acetate 0.1% NP-40, 500 µM TCEP, 10 mM TSY, 10 mM PCA, 12-fold diluted PCD (Pacific Biosciences), and 200-fold diluted RNasin® Plus RNase Inhibitor (Promega). 5 µg/ml yeast RNA was included for the contribution test of exogenous RNA to the oligomer formation of Vasa. The fluorescent images illuminated by a 532-nm laser (Coherent, Sapphire) were acquired at a frame rate of 100 ms using an EMCCD camera (Andor, iXon+) equipped with a Nikon Ti-E TIRF microscope[19].

**Aggregation measurement of Vasa.** In the Vasa assembly experiment in the presence of RNA, the plasma-cleaned glass surface of the microchamber was coated with amino-silane and PEG-biotin[19]. Briefly, 1 mg/ml Neutralised Avidin (Wako) was adsorbed onto the glass surface, and then the excess avidin was removed by washing three times with buffer after 2 min incubation. Biotinylated TMR-labelled Vasa, labelled with Biotin-XX-NHS (Sigma-Aldrich), was fixed on the substrate. After introducing 5 µg/ml yeast RNA and removing the excess, 50 nM TMR-Vasa was applied during the recording, and a series of 1000 fluorescent images were acquired at a time interval of 600 ms in total using TIRF microscopy.

**Image analysis of oligomerized Vasa.** The fluorescence intensities of TMR-labelled Vasa were analysed, and the size of the optimal region of interest (ROI) was 8 × 8 pixels (scale size of 108 nm/pixel) and the noise tolerance was ~1500 using the plugin of spot intensity analysis (FIJI, ImageJ; NIH).

To determine the fluorescence intensity of a single TMR molecule, we acquired successive ~1000 images of TMR-Vasa in the absence of oxygen scavengers and then analysed each bleaching transition of fluorescence intensities by a hidden Markov Model using the MATLAB program (MathWorks, Fig. S1a)[20]. The intensity of a single TMR was determined from the averaged stepwise intensity (178.2 ± 58.6, mean ± SD) (Fig. S1b). We determined the number of oligomerized Vasa by approximating the measured intensity as the single TMR intensity multiplied by the number of molecules.

**Fluorescent measurement of target RNA.** For the measurement of red fluorescence intensity only, we mixed 20 nM target RNA with 760 nM Siwi for 20 min (endogenous target RNA) or 3 h (exogenous piRNA4 target RNA) at room temperature, and then fixed Siwi-piRISC on the glass surface via biotinylated target RNA by avidin-biotin binding. We monitored the fluorescence intensity of target RNA illuminated by a 642-nm laser (Coherent, cube) soon after adding 25 nM non-fluorescent target RNA and ATP in the assay buffer, including oxygen scavengers. We monitored 1000 successive images each in the time interval of 400 ms. The image analysis of single molecular tracking of red fluorescence spots was the same as explained above.

**Two (or three)-colour measurement of target RNA, Vasa and/or Siwi.** For the simultaneous measurement of cleaved target RNA and vasa, a premix of 10 nM biotinylated target RNA (labelled with ATTO647N), 625 nM vasa (labelled with TMR), and 380 nM Siwi (antibody labelled with Alexa488, illuminated by a 488-nm laser, Coherent, cube) and was fixed on the surface by avidin-biotin binding in the assay buffer containing the oxygen scavenger. To measure the dissociation of cleaved target RNA and Vasa, 1 mM ATP was introduced into the chamber during recording. We acquired 1000 successive images each in a time interval of 400 ms by shutter control. The image analysis of single molecular tracking was the same as that explained above.

**Statistics and reproducibility.** Statistical analyses were all performed using MATLAB 2020b and Curve Fitting Toolbox. The averaged intensity of a single fluorescent dye was analysed by fitting data with a single Gaussian curve (Figs. S1b, S3d). The estimated number of molecules represents the integer value obtained by dividing the measured intensity by the averaged single-molecule intensity (Figs. 1b, d, 3b–e, S3). The intensity in association or dissociation were analysed with a

hidden Markov Model (Figs. 1d, 3c–e, S5). Distributions or histograms were analysed by fitting data with a single exponential curve (Figs. 1e–f, 2c, e). The oligomerized number of Vasa was independent of the dwell time evaluated by Welch's t-test (p value > 0.05, Fig. S1c).

**Reporting summary**. Further information on research design is available in the Nature Research Reporting Summary linked to this article.

## Data availability

The authors declare that the data are available within the paper and its Supplementary information files.

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

## Acknowledgements

We would like to thank M. C. Siomi for discussing this work. We also thank T. Kinoshita, K. Sakakibara, and all other members of the M. C. Siomi laboratory and S. Uemura laboratory for technical support and discussion. This work was supported by research grants from JST-CREST (JPMJCR14W1).

## Author contributions

Y.K. performed the fluorescent experiments and data analyses. Y.K., R.M., and N.M. purified the proteins and performed biochemical assays and immunostaining experiments. S.K. performed the simulations. Y.K., R.I., and S.U. wrote the manuscript.

## Competing interests

The authors declare no competing interests.
