## [Peer Review File · Communications Biology]

Reviewers' comments:

Reviewer #1 (Remarks to the Author):

The manuscript by Kinoshita et al uses single-molecule techniques to analyze various functions of the DEAD-box helicase Vasa. The experimental data are in my view technically sound. However, serious problems with the interpretation of the data need to be addressed before publication.

Much of the paper is based on comparing the behaviour of full-length Vasa with that of a N-terminally deleted form (dN-Vasa). dN-Vasa lacks a disordered region that is characteristic of Vasa and many other DEAD-box helicases. In Figure 1, the authors show that dN-Vasa, unlike full-length Vasa, fails to oligomerize and to form Nuage. From this they conclude that Vasa dimerization is required for Nuage formation. The problem is that this experiment does not prove causality. dN-Vasa could be deficient in some activity other than dimerization that results in an inability to form Nuage. A gain-of-function experiment involving dimerizing dN-Vasa through some other means and thereby rescuing Nuage formation would be necessary for this conclusion to be proven. A similar logical problem affects the interpretation of Figure 2 regarding the potential role of Vasa dimerization on RNA unwinding activity. Lacking additional experimental data of the nature suggested, the authors should be more circumspect in their conclusions.

On page 5 the authors state that the degree of Vasa oligomerization in the presence of Siwi-piRISC was 'almost consistent' with that of Vasa alone, and that there was 'little difference' with and without Siwi-piRISC. This is simply not true. In Fig. 1b (Vasa alone), less than 10% is found in oligomers larger than dimers, while in Fig. 3b around 90% is found in oligomers larger than dimers, with 15-20% in octamers or larger. The description of these experiments needs to be made accurate, and effects of Siwi-piRISC incorporated in the discussion.

Minor comments:

In the paragraph on page 5 entitled 'Colocalization...', the first paragraph is very unclear. What are 'piRNAs biased with 1U'? The next sentence also requires rewriting as I cannot figure out exactly what Siwi-piRISC components are included in the assay.

On page 7, third line from bottom, what is meant by calling DDX43 'the same type of Vasa recently identified'?

In the results section, dimerization and oligomerization are often used interchangeably, and this is confusing, particularly in the context of my comment above about differences with and without Siwi-piRISC.

On page 3, line 4, dimerization is misspelled.

Reviewer #2 (Remarks to the Author):

The germline small RNA pathway consisting of Piwi proteins and Piwi-interacting RNAs are essential for transposon silencing in animal germlines. One of the factors essential for piRNA biogenesis is a DEAD box RNA helicase called Vasa (DDX4). Biogenesis of piRNAs is initiated by cleavage of a target RNA by the piRNA-guided Piwi proteins. Once cleaved, one of the cleavage fragments is transferred to a new Piwi protein where it is matured as a new piRNA. Vasa is implicated in this release of the cleavage fragment. The authors use single-molecule analysis of Vasa to show that dimerization of Vasa is essential for this release, *in vitro* and *in vivo*. This study provides new and interesting insights into the molecular events taking place immediately after Piwi slicing of the target RNA. It should be of great interest to the small RNA community, hence should be published.

Comments

Fig. 1b shows that Vasa on its own form mainly dimers, but when it binds to the RNA via Siwi complex, it is more random, but clearly starts as a dimer (Fig. 3b). I don't understand why the

authors say in line 166 "there was little difference in the oligomerization of Vasa...in these complexes and Vasa itself". May this needs to be clarified better in the text.

Fig3c. Why is the RNA signal not at 1 (Y-axis on the right of the panel)? Provide a brief explanation in the legend or main text.

What promotes the dimerization of Vasa? Did the authors do experiments with different N-terminal region truncations? At least, briefly discuss the possibilities.

The model would be that Vasa exists mostly as dimers in solution, this is mediated via N-term, associates with the Siwi-target RNA complex and then dissociates simultaneously after the cleaved fragment is also released. This is the simple situation. In more complex situations, the dissociations are unsynchronized, or with Vasa leaving as monomers etc. This latter part could be due to the quality of the Siwi-piRNA complex that is being used. Since these are endogenous Siwi proteins that are loaded with additional guide RNAs, it could be a very heterogeneous mix. Do we know how much of loading is there? If we one were to sequence the RNAs in the piRNA4-loaded Siwi complex, how much of the reads will be from piRNA4? This is not essential, but if the authors have, it could be mentioned.

Reviewer #3 (Remarks to the Author):

In the manuscript, Kinoshita et al. apply the use of single molecule imaging to show that the N-terminal extension of BmVasa, a germ-specific DEAD box RNA helicase, is necessary for the protein oligomerization and for the dissociation of the target RNA from the Siwi-piRISC complex.

Overall, the manuscript is clear and well written, the results are exhaustively discussed, and the use of single-molecule fluorescence techniques seems appropriate to address unresolved questions about BmVasa mode of action.

I would have only few comments:

- Does RNase A treatment of lysates before BmVasa purification affect oligomerization?
- What is the ATPase activity and RNA binding affinity of dN-Vasa as compared to WT-Vasa?

Minor corrections:

Line 76: ddimerization typo

Line 22: add "may contain" as not all C-/N- terminal extensions of RNA helicases have IDRs

Reviewers' comments:

Reviewer #1 (Remarks to the Author):

The manuscript by Kinoshita et al uses single-molecule techniques to analyze various functions of the DEAD-box helicase Vasa. The experimental data are in my view technically sound. However, serious problems with the interpretation of the data need to be addressed before publication.

Thank you very much for your many valuable comments. We have responded below to all of them carefully.

Much of the paper is based on comparing the behaviour of full-length Vasa with that of a N-terminally deleted form (dN-Vasa). dN-Vasa lacks a disordered region that is characteristic of Vasa and many other DEAD-box helicases. In Figure 1, the authors show that dN-Vasa, unlike full-length Vasa, fails to oligomerize and to form Nuage. From this they conclude that Vasa dimerization is required for Nuage formation. The problem is that this experiment does not prove causality. dN-Vasa could be deficient in some activity other than dimerization that results in an inability to form Nuage. A gain-of-function experiment involving dimerizing dN-Vasa through some other means and thereby rescuing Nuage formation would be necessary for this conclusion to be proven. A similar logical problem affects the interpretation of Figure 2 regarding the potential role of Vasa dimerization on RNA unwinding activity. Lacking additional experimental data of the nature suggested, the authors should be more circumspect in their conclusions.

We totally agree with the reviewer 1's assessment. As you pointed out, we agree that this conclusion was an overstatement, therefore we have tone downed as follows.

- 1. Changed from "essential" to "preferable" on page 1, line 32, on page 3, line 4 and on page 29, line 12.**
- 2. Changed to "N-terminal IDR contribution for Vasa's dimerization" on page 3, line 10.**
- 4. Changed to "suggesting that Vasa's dimerization at N-IDR triggers Nuage formation *in vivo*." on page 3, line 24.**
- 5. Changed from "is required for" to "is preferable for" on page 7, line 32.**

We also agree on the interpretation issue you pointed out about the potential role of the dimerization on unwinding activity. According to first sentence on page 3, line 28, exogenous RNA may contribute Vasa's aggregation through recognition sites other than IDRs.

To investigate the contribution of an exogenous RNA, we have performed the additional experiments and included results of oligomerization in the presence and absence of exogenous (yeast) RNA as the stripe bar in Fig. 1b. The result demonstrates that the exogenous RNA contributes to the further oligomerization of Vasa dimerized through IDR. Without these results, we would not have been able to explain why dN-Vasa retains some unwinding activity (Fig. 2c). However, with these results, we are now able to explain why the cleavage assay performed in the presence of exogenous RNA showed a slight unwinding activity due to the formation of some amounts of dimeric dN-Vasa.

Based on these results, we have majorly changed sentences related these points on page 3, line 33 and on page 4, line 7.

On page 5 the authors state that the degree of Vasa oligomerization in the presence of Siwi-piRISC was 'almost consistent' with that of Vasa alone, and that there was 'little difference' with and without Siwi-piRISC. This is simply not true. In Fig. 1b (Vasa alone), less than 10% is found in oligomers larger than dimers, while in Fig. 3b around 90% is found in oligomers larger than dimers, with 15-20% in octamers or larger. The description of these experiments needs to be made accurate, and effects of Siwi-piRISC incorporated in the discussion.

Thank you for pointing this out. We fully agree with your point and added a few sentences in this paragraph on page 5, line 25.

Minor comments:

In the paragraph on page 5 entitled 'Colocalization...', the first paragraph is very unclear. What are 'piRNAs biased with 1U'? The next sentence also requires rewriting as I cannot figure out exactly what Siwi-piRISC components are included in the assay.

As you have pointed this out, we have deleted the first paragraph on page 5 and added the sentence in Materials and Methods on page 10, line 33.

On page 7, third line from bottom, what is meant by calling DDX43 'the same type of Vasa recently identified'?

Thank you for your comments. We have changed the words "the same type of Vasa" to "one of DEAD-box RNA helicase" on page 8, line 1.

In the results section, dimerization and oligomerization are often used interchangeably, and

this is confusing, particularly in the context of my comment above about differences with and without Siwi-piRISC.

We agree with your point about the ambiguity in the definition of how the two terms “Dimerization” and “Oligomerization” are used in the results section. We have clarified as follows, “Dimerization” is based on the dimerization through only IDRs, in contrast “Oligomerization” is based on the oligomerization through either exogenous RNA or Siwi-piRISC or both. These terms have been standardized according to these criteria in the whole text.

On page 3, line 4, dimerization is misspelled.

Thank you for pointing this out. We decided that "oligomeraization" was more suitable than "dimerization" based on above criteria.

Reviewer #2 (Remarks to the Author):

The germline small RNA pathway consisting of Piwi proteins and Piwi-interacting RNAs are essential for transposon silencing in animal germlines. One of the factors essential for piRNA biogenesis is a DEAD box RNA helicase called Vasa (DDX4). Biogenesis of piRNAs is initiated by cleavage of a target RNA by the piRNA-guided Piwi proteins. Once cleaved, one of the cleavage fragment is transferred to a new Piwi protein where it is matured as a new piRNA. Vasa is implicated in this release of the cleavage fragment. The authors use single-molecule analysis of Vasa to show that dimerization of Vasa is essential for this release, in vitro and in vivo. This study provides new and interesting insights into the molecular events taking place immediately after Piwi slicing of the target RNA. It should be of great interest to the small RNA community, hence should be published.

Thank you for your positive comments.

Comments

Fig. 1b shows that Vasa on its own form mainly dimers, but when it binds to the RNA via Siwi complex, it is more random, but clearly starts as a dimer (Fig. 3b). I don't understand why the authors say in line 166 “there was little difference in the oligomerization of Vasa...in these complexes and Vasa itself”. May this needs to be clarified better in the text.

Thank you for pointing this out. We agree that this is a reasonable point, since Reviewer 1 also raised the same point. We have corrected a few sentences in this paragraph on page 5, line 25.

Fig3c. Why is the RNA signal not at 1 (Y-axis on the right of the panel)? Provide a brief explanation in the legend or main text.

Thank you for your points, we have changed the label of Number of molecules to "Estimated number of molecules" and added one sentence for the explanation of the calculation of molecules in Figure legends in Fig 3, S3, and S5.

What promotes the dimerization of Vasa? Did the authors do experiments with different N-terminal region truncations? At least, briefly discuss the possibilities.

Thank you for your important question. We believe that Vasa dimerization is enhanced by the interaction between IDRs. We have not performed experiments using the mutant with different truncated regions. By focusing on the repetitive regions in the IDRs, experiments using various truncated regions will reveal more details about the required regions for dimerization. The sentence has been added on page 7, line 7.

The model would be that Vasa exists mostly as dimers in solution, this is mediated via N-term, associates with the Siwi-target RNA complex and then dissociates simultaneously after the cleaved fragment is also released. This is the simple situation. In more complex situations, the dissociations are unsynchronized, or with Vasa leaving as monomers etc. This latter part could be due to the quality of the Siwi-piRNA complex that is being used. Since these are endogenous Siwi proteins that are loaded with additional guide RNAs, it could be a very heterogeneous mix. Do we know how much of loading is there? If we one were to sequence the RNAs in the piRNA4-loaded Siwi complex, how much of the reads will be from piRNA4? This is not essential, but if the authors have, it could be mentioned.

Thank you for your good question. The data in Fig. S4, demonstrating Complex formation of Siwi-piRISC with RNA and Vasa may offer an answer: by calculating the number of molecules for which the fluorescence of the Siwi antibody co-localizes that of the RNA, we could obtain an approximate efficiency (less than 10%) in the piRNA4-loaded Siwi complex formation.

Reviewer #3 (Remarks to the Author):

In the manuscript, Kinoshita et al. apply the use of single molecule imaging to show that the N-terminal extension of BmVasa, a germ-specific DEAD box RNA helicase, is necessary for

the protein oligomerization and for the dissociation of the target RNA from the Siwi-piRISC complex.

Overall, the manuscript is clear and well written, the results are exhaustively discussed, and the use of single-molecule fluorescence techniques seems appropriate to address unresolved questions about BmVasa mode of action.

Thank you for your positive comments.

I would have only few comments:

- Does RNase A treatment of lysates before BmVasa purification affect oligomerization?

You raised good points. We have immunoprecipitated the lysate after RNase treatment to confirm number of Vasa molecules by electrophoresis, which resulted in the formation of at least a dimer (unpublished data).

- What is the ATPase activity and RNA binding affinity of dN-Vasa as compared to WT-Vasa?

Thank you for your good question. In the experiments performed in Refs. 4 and 7, the ATPase activity and RNA-binding ability were confirmed using a construct similar to dN-Vasa, although the comparison with WT was not observed, therefore it is expected that the N terminus is not directly affecting the enzyme activity itself.

Minor corrections:

Line 76: ddimerization typo

Thank you for pointing this out. We decided that "oligomeraization" was more suitable than "dimerization" in this case.

Line 22: add "may contain" as not all C-/N- terminal extensions of RNA helicases have IDRs

Thank you for pointing this out. We have added "may" just before "contain" on page 1, line 22.

REVIEWERS' COMMENTS:

Reviewer #1 (Remarks to the Author):

The revised version of this manuscript largely addresses my concerns. However, the change of wording from 'essential' or 'is required for' to 'preferable' may be confusing to some readers. The point is that deletion of the IDR abrogates dimerization as well as nuage formation and RNA unwinding activity. The simplest interpretation is indeed that dimerization is required for these activities. However, it needs to be made explicit that it cannot be ruled out that the IDR has functions other than promoting dimerization that are essential for these activities.

Aside from that, 'exogenous' is misspelled on page 4, line 10.

Reviewer #3 (Remarks to the Author):

Authors have addressed my previous comments.